# Diagnostic Performance of Different Laboratory Methods for the Detection of Extrapulmonary Tuberculosis

**DOI:** 10.3390/microorganisms11041066

**Published:** 2023-04-19

**Authors:** Mohammad Khaja Mafij Uddin, Md. Fahim Ather, Senjuti Kabir, Arfatur Rahman, Sabrina Choudhury, Rumana Nasrin, Tanjina Rahman, S. M. Mazidur Rahman, Shahriar Ahmed, Sayera Banu

**Affiliations:** 1Infectious Diseases Division, International Centre for Diarrhoeal Disease Research, Bangladesh, Dhaka 1212, Bangladesh; kmuddin@icddrb.org (M.K.M.U.);; 2Monash Institute of Pharmaceutical Sciences, Monash University (Parkville Campus), 381 Royal Parade, Parkville, VIC 3052, Australia

**Keywords:** tuberculosis, extrapulmonary, diagnostic, GeneXpert, sensitivity, specificity

## Abstract

Accurate and appropriate extrapulmonary tuberculosis (EPTB) diagnosis remains challenging due to its paucibacillary nature, requirement of invasive collection procedures, and lack of sensitive tests. This study investigated the diagnostic performance of different methods for the diagnosis of EPTB. A total of 1340 EPTB specimens were collected from presumptive EPTB patients from four different hospitals between November 2015 and March 2017. The collected specimens were tested with AFB microscopy, culture, Xpert MTB/RIF assay (Xpert), and MTBDR*plus* assay. Among the 1340 EPTB specimens, 49 (3.66%), 141 (10.52%), 166 (12.39%), and 154 (11.49%) were positive in AFB microscopy, culture, Xpert MTB/RIF, and MTBDR*plus* assay, respectively. A total of 194 (14.9%) cases were found positive in at least one of these methods. Using culture as a reference standard, the sensitivity and specificity of AFB microscopy, Xpert MTB/RIF, and MTBDR*plus* assay were: 27.0%/99.1%, 83.7%/96.0%, and 79.4%/96.5%, respectively. Compared to the composite reference standard, the sensitivity of culture, AFB microscopy, Xpert MTB/RIF, and MTBDR*plus* assay was 72.7%, 25.3%, 85.6%, and 79.4%, respectively, with a specificity of 100% for all the methods. The Xpert MTB/RIF assay showed the highest sensitivity compared to other methods. Considering the short turnaround time and promising findings, Xpert MTB/RIF assay should be integrated into national TB guidelines as a routine diagnostic test.

## 1. Introduction

Extrapulmonary tuberculosis (EPTB) is considered to be a secondary manifestation of *Mycobacterium tuberculosis* complex (MTBC) infection, which causes tuberculosis (TB) in organs other than the lungs. It is an emerging infection that leads to a high incidence of morbidity and mortality. According to the Global Tuberculosis Report 2022, 17% of 10.6 million TB cases in 2021 were detected as EPTB [1]. The burden of EPTB is underestimated due to the similitude with some other diseases, the paucibacillary nature, low infection potential, and technical diagnostic difficulties [2,3]. EPTB can affect any organ of the body, but it usually involves lymph nodes, bones, the central nervous system, the gastrointestinal system, the genitourinary system, etc. [4,5]. Its clinical resemblance with other diseases confuses physicians for cytological or histopathological confirmation, which causes ambiguity with the accuracy of diagnosis of EPTB [3]. Some conditions or infections can present similar cytology or histopathology to TB lymphadenopathies, such as some fungal infections, inflammatory conditions, or sarcoidosis [6], which leads to over- or under-diagnosis. Diagnosing EPTB using histopathological or clinical methods may be possible, but microbiological identification poses a great challenge to TB management globally. One of the major barriers to microbiological identification of EPTB is the difficulty of obtaining specimens from infection sites, which involves invasive surgery in many cases. Moreover, the lack of appropriate instruments or laboratory support in the peripheral regions of the country is an added challenge [4].

Microscopic detection through acid-fast bacillus (AFB) staining is the simplest and most widely used laboratory diagnostic method for TB. However, the paucibacillary nature of the EPTB specimen yields very few bacilli and, consequently, is associated with low sensitivity, with around 10% positivity in many cases [6]. On the other hand, mycobacterial culture is considered the “gold standard” of TB diagnosis; however, positivity for EPTB varies among specimen types [5,6]. Polymerase chain reaction (PCR) targeting different conserved sequences (e.g., IS*6110*, IS*1081* 65 kDa, GCRS, etc.) in the MTBC genome has been used for the diagnosis of both pulmonary TB (PTB) and EPTB specimens, including stool-based PCR [6,7,8,9,10,11,12]. The presence of PCR inhibitors in the clinical specimens can compromise the diagnostic performance of this technique [13]. In the last decade, significant advances have been made in TB diagnostics, but there is still a need to validate the diagnostic yield of conventional culture with rapid-molecular methods such as Xpert MTB/RIF and GenoType MTBDR*plus* in EPTB specimens. Xpert MTB/RIF is a semi-nested real-time quantitative PCR (RT-qPCR)-based assay that targets the MTBC-specific rpo*B* gene for simultaneous detection of MTBC and Rifampicin (RIF) susceptibility [14]. GenoType MTBDR*plus* (MTBDR*plus*) is a DNA probe hybridization-based assay that can simultaneously detect MTBC, and most of the common mutations are associated with RIF and isoniazid (INH) resistance. Both molecular tests have excellent sensitivity and specificity for the detection of MTBC and drug resistance, which have been approved by the World Health Organization (WHO). Many studies have been conducted on these two molecular methods for various EPTB specimens from diverse geographical settings [2,5,15,16]. Bangladesh is a high TB and MDR-TB burden country. Different conventional TB detection methods like bright-field (BF) microscopy, light-emitting diode (LED) microscopy, and Lowenstein–Jensen (LJ) culture are being used for the diagnosis of TB in Bangladesh [17]. The National TB Control Programme (NTP) has rolled out Xpert MTB/RIF across the country for the detection of TB and MDR-TB among PTB patients.

Though the diagnostic performance of these conventional and molecular techniques on EPTB specimens has been reported from different geographical settings globally, such reports from Bangladesh are very limited. To our knowledge, only one previous study from Bangladesh has compared the diagnostic performance of microscopy and LJ culture for EPTB diagnosis [17]. Better understanding is required to facilitate standardization and optimal preparation of samples as well as the role of diagnostic methods for bacteriological confirmation of EPTB in different settings. In this study, we aimed to investigate the diagnostic performance of conventional and rapid-molecular methods for the detection of EPTB from diverse types of presumptive EPTB specimens.

## 2. Materials and Methods

### 2.1. Study Settings and Participants

The study protocol was reviewed and approved by the Research Review Committee and the Ethical Review Committee of icddr,b. This cross-sectional study was conducted between November 2015 and March 2017 in four tertiary healthcare facilities in Dhaka-Dhaka Medical College Hospital (DMCH), Sir Salimullah Medical College and Mitford Hospital (SSMCH), Shaheed Suhrawardy Medical College and Hospital (ShSMCH), and Dhaka National Medical College and Hospital (DNMCH). All these hospitals are well equipped with expert physicians, and, every day, more than 800 people, either in-patients or out-patients, visited each of these hospitals. In this study, presumptive EPTB patients were identified by the local treating physicians among the patients admitted to these four hospitals. After taking informed written consent from patients (>18 years old), from parents/caregivers (<10 years old), and assent from both patients and their parents/caregivers (10≤ 18 years old), the study staff collected sociodemographic and clinical data from enrolled patients. Patients with PTB or those patients’ other serious co-morbid conditions (chronic obstructive pulmonary disease, chronic kidney disease, acute renal failure, chronic liver disease, any heart condition, etc.), on anti-TB treatment, and who did not consent, were excluded.

### 2.2. Specimen Processing, Microscopy and Culture

A total of 1340 specimens from presumptive EPTB patients were collected by local treating physicians as per their standard practice and transported to the Mycobacteriology Laboratory of icddr,b on the same day, maintaining an appropriate cold chain. At the laboratory, some specimens like urine and pleural fluid were concentrated by centrifugation before digestion and decontamination. Tissue specimens were homogenized using a homogenizer before processing. Each specimen was divided into two equal parts: one part for Xpert MTB/RIF assay and another part, of all specimens, except cerebrospinal fluid (CSF), were digested and decontaminated by N-acetyl-L-cysteine (NALC)-Sodium hydroxide (NaOH) method as previously described [18]. The processed specimen was then used for AFB microscopy, solid culture, and MTBDR*plus* assay.

### 2.3. Xpert MTB/RIF Assay

The Xpert MTB/RIF assay was performed according to the manufacturer’s instructions [19]. In brief, the sample reagent was added to untreated or centrifuged specimens in a 2:1 ratio, mixed, and incubated for 15 min at room temperature. Then, 2 mL of processed specimens were transferred to Xpert MTB/RIF Cartridge (Version 4.0), the cartridge was loaded into the Xpert MTB/RIF machine, and software-generated results were recorded.

### 2.4. GenoType MTBDRplus Assay

The GenoType MTBDR*plus* assay (Hain Life Sciences, Nehren, Germany, Version 2.0) was performed on processed specimens according to the manufacturer’s instructions [20]. DNA was extracted using the genolyse kit, multiplex PCR amplification was performed, and the PCR products were hybridized with specific probes coated on a membrane strip. The susceptibility patterns were identified using the interpretation sheet provided by the manufacturer.

### 2.5. Statistical Analysis

Data were entered and analyzed using the Statistical Package for the Social Sciences software (SPSS) version 20.0. The sensitivity, specificity, and predictive values with 95% confidence intervals (CIs) were calculated to assess the diagnostic performance of different tests for EPTB diagnosis. McNemar’s test was used for the comparative analysis, where *p*-value < 0.05 was considered to be statistically significant. The area under the receiver operating characteristic (ROC) curve (AUC) was used to evaluate the overall performances of all the diagnostic methods compared to the composite reference standard (CRS) using RStudio version 1.4. The CRS-positive was considered for confirmed positive cases in any of AFB microscopy, culture, and molecular methods (Xpert MTB/RIF and MTBDR*plus*) [21]. The AUC with >0.5 to <0.7 demonstrated a low diagnostic value, ≥0.7 to <0.9 demonstrated a moderate diagnostic value, and ≥0.9 demonstrated a high diagnostic value [22].

## 3. Results

### 3.1. Type of EPTB Specimens and Bacteriologically Positive Cases

In this study, we enrolled a total of 1340 presumptive EPTB patients with diverse specimen types. All the enrolled specimens were categorized into several broad categories, as shown in Figure 1. Among them were pleural fluids (n = 318, 28.43%), ascitic fluids (n = 244, 18.21%), pus (n = 177, 13.21%), CSF (n = 156, 11.64%), tissues (n = 116, 8.66%), urine (n = 122, 9.1%), fine needle aspirates (FNAC) (n = 106, 7.91%), and synovial fluids (n = 38, 2.84%). A total of 194 (14.48%) cases were confirmed as bacteriologically TB-positive from all the testing methods mentioned above. Of the 194 positive cases, pleural fluids, ascitic fluids, pus, CSF, tissues, urine, FNAC, and synovial fluids were 18 (9.28%), 16 (8.26%), 66 (34.02%), 13 (6.7%), 34 (17.52%), 6 (3.09%), 40 (20.62%), and 1 (0.51%), respectively (Figure 1).

### 3.2. Overall Performance of Four Laboratory Methods in Detecting EPTB Cases

Among the 1340 enrolled cases, 49 (3.66%) and 141 (10.52%) were positive in AFB microscopy and culture, respectively, while the yields of two molecular methods, Xpert MTB/RIF and MTBDRplus, were 12.39% (166) and 11.49% (154), respectively (Table 1). Diagnostic performances of AFB microscopy, Xpert MTB/RIF, and MTBDRplus assays among the 1340 cases were evaluated against the culture method. AFB microscopy had the lowest sensitivity (27.1%, 95% CI 20–35.3) and accuracy (91.4%) among the three methods. The two molecular methods, Xpert MTB/RIF and MTBDRplus, showed a similar level of sensitivities and the exact level of accuracy. The sensitivities of Xpert MTB/RIF and MTBDRplus assays were 83.7% (95% CI: 76.5–89.4) and 79.4% (95% CI: 71.8–85.8), respectively, against culture, while the accuracy was 94.7% each. However, the specificity of AFB microscopy was higher than the Xpert MTB/RIF and MTBDRplus assays when compared to culture, which was valued by 99.1% (95% CI: 98.4–99.5), 96.0% (95% CI: 94.7–97.0), and 96.5% (95% CI: 95.3–97.5), respectively (Table 1).

Additionally, the diagnostic performances of culture, AFB microscopy, Xpert MTB/RIF, and MTBDRplus assays were evaluated against CRS (Table 2). The Xpert MTB/RIF assay showed the highest sensitivity of 85.6% (95% CI: 79.8–90.2) and accuracy of 97.6% against CRS, followed by the MTBDRplus, culture, and AFB microscopy methods. The sensitivities of MTBDRplus, culture, and AFB microscopy were 79.4% (95% CI: 73.0–84.8), 72.7% (95% CI: 65.8–78.8), and 25.3% (95% CI: 19.3–32.0), respectively, while the accuracies were 97%, 96%, and 89.2%, respectively. The specificities for all four methods were 100% (99.7–100) (Table 2). The ROC curve was prepared for the evaluation of the diagnostic performances for all four diagnostic methods. Considering culture as the reference, the values of AUC for AFB microscopy, Xpert MTB/RIF, and MTBDRplus were 0.630, 0.898, and 0.880, respectively. Also, considering the consensus standard as the reference, the AUC values for culture, AFB microscopy, Xpert MTB/RIF, and MTBDRplus assays accounted for 0.863, 0.626, 0.928, and 0.870, respectively (Table 1 and Table 2; Figure 2).

### 3.3. Diagnostic Performance of Four Diagnostic Methods for the Detection of EPTB Cases in Various Specimen Types

The sensitivity, specificity, and accuracy of all four diagnostic methods for different EPTB specimens were evaluated compared to CRS (Figure 3). Xpert MTB/RIF assay showed the highest ranges of sensitivity across all the EPTB specimens with 62.5–100%, which was followed by both culture and MTBDRplus assay with 50.0–100% each, while AFB microscopy showed the lowest range of sensitivity of 5.6–100%. Similarly, the accuracies of Xpert MTB/RIF, MTBDRplus assay, culture, and AFB microscopy ranged from 95.3–100%, 90.6–100%, 87.9–100%, and 69.8–100%, respectively. All the methods except AFB microscopy showed more than 96% accuracy for pleural fluids, ascitic fluids, CSF, and urine for EPTB detection. Additionally, the Xpert MTB/RIF method showed exactly 98.3% accuracy for both pus and tissue specimens, while MTBDRplus assay and culture had 97.7% and 91.1% accuracies for pus and 94.8% and 87.9% for tissue specimens, respectively. A high-to-moderate range of accuracy was observed in terms of FNACs against Xpert MTB/RIF, culture, MTBDRplus assay, and AFB microscopy, which was 95.3%, 92.5%, 90.6%, and 69.8%, respectively. All four methods revealed 100% accuracy and sensitivity for synovial fluid specimens (Figure 3).

### 3.4. Semi-Quantitative Distribution and Detection by AFB Microscopy and Xpert MTB/RIF Assay

The semi-quantitative assessment of bacterial load in the EPTB specimens was investigated utilizing AFB microscopy and Xpert MTB/RIF assays. From the 49 AFB microscopy-positive cases, 21 (42.9%) were found scanty and 18 (36.7%) were 1+, while both the 2+ and 3+ categories had 5 (10.2%) cases each. Pus (31) was the highest-yielding specimen in AFB microscopy, where 10, 14, 3, and 4 specimens were scanty, 1+, 2+, and 3+ categories, respectively. This was followed by FNAC, tissue, and urine, where 8, 5, and 3 specimens were positive in AFB microscopy with different gradings, respectively. Pleural fluid and synovial fluid found only 1 positive case each with scanty grading, while no positive cases were found in terms of ascitic fluid and CSF (Table 3).

In terms of Xpert MTB/RIF-positive cases, the highest proportion of specimens was of low burden (n = 76, 45.8%), which was followed by very low, medium, and high categories with (n = 54, 32.5%), (n = 28, 16.9%), and (n = 8, 4.8%) samples, respectively. Similar to AFB microscopy, most EPTB-positive cases were found from pus specimens (63) in the Xpert MTB/RIF test, where 8, 31, 19, and 5 were very low, low, medium, and high categories. The majority of the pus (n = 31)-, FNAC (n = 18)-, and pleural fluid (n = 7)-positive cases were low categories, while 15 tissue specimens were from very low and low categories each. All the ascitic fluid-positive cases (n = 10) were detected as very low, while only 1 synovial fluid had a medium level of bacteria (Table 3).

### 3.5. Comparison of Two Rapid Molecular Methods for Detection of EPTB and Rifampicin Susceptibility

A total of 166 and 154 specimens were found to be positive for EPTB, respectively, by Xpert MTB/RIF and MTBDRplus assay. These two methods demonstrated 97.8% accuracy with a moderate level of sensitivity (87.3%, 95% CI: 81.3–92.0) for EPTB detection among them (Table 4). The susceptibility patterns determined by these two tests were compared since both tests can identify RIF susceptibility. A total of 145 cases were conclusive to determine the RIF susceptibility by both methods. Among them, 143 cases were RIF-sensitive and 2 cases were RIF-resistant in both methods, revealing 100% sensitivity and specificity of MTBDRplus assay compared to the Xpert MTB/RIF assay (Table 4).

## 4. Discussion

In this study, we assessed the diagnostic performance of two conventional (AFB microscopy and LJ culture) and two rapid-molecular (Xpert MTB/RIF and GenoType MTBDR*plus*) TB detection methods on 1340 presumptive EPTB cases. Among them, 194 cases were positive in any of the diagnostic methods and considered CRS-positive. Though the culture test has always been considered the gold standard for TB diagnosis, it is not ideal for EPTB detection. In terms of paucibacillary cases such as EPTB, the culture yield might be poor [23,24]. However, advanced molecular methods can detect the DNA from the nonviable bacilli even with low bacterial burden and hence can identify culture-negative TB cases. To overcome such issues regarding the performance of different diagnostic methods on EPTB, both culture and CRS have been used as reference standards in this case, like other studies, to reach an optimum sensitivity, specificity, and accuracy of the different diagnostic methods for EPTB detection [24,25].

Our study found that the Xpert MTB/RIF assay was the highest-yielding diagnostic method for EPTB detection with pooled sensitivities of 83.7% and 85.6% when compared with culture and CRS, respectively, whereas the specificities were 96.0% and 100%, respectively. Several studies have reported similar sensitivity of Xpert MTB/RIF assay ranging from 87.3% to 95% and the specificity over 97% in contrast to CRS [26,27]. MTBDR*plus* followed the Xpert MTB/RIF assay in diagnosing the EPTB cases in this investigation. The pooled sensitivity of MTBDR*plus* was 79.4% against both the culture and CRS, whereas the specificities were 96.5% and 100%, respectively. Effective data on the evaluation of MTBDR*plus* assay against EPTB is scarce; however, a recent study conducted by Singh, B.K. et al. reported similar sensitivity (73.4%) of MTBDR*plus* compared to liquid culture (MGIT) [16]. In contrast to these molecular methods, the conventional AFB microscopy showed the lowest-yielding performance for the detection of EPTB cases with a sensitivity of 27.0% and 25.3% against culture and CRS, respectively, whereas the specificity was 99.1% and 100%, respectively. This poor sensitivity of AFB microscopy, however, is not surprising, as former studies have already described the lower sensitivity of this model [28,29]. The specificity of AFB microscopy (99.1%) is higher in terms of culture compared to the Xpert MTB/RIF (96%) and MTBDR*plus* assay (96.5%). This is due to the missing of paucibacillary cases by the culture, which turned out to be positive in the molecular methods and was considered false-positive (classified as positive by the molecular methods and negative by culture) [16].

Along with the sensitivity, specificity, and accuracy of certain diagnostic methods, the ROC curve analysis was also investigated in this study to get a clear idea of their diagnostic performances. In comparison to culture, the AUC of AFB microscopy, Xpert MTB/RIF, and MTBDR*plus* assays were constituted of 0.630, 0.898, and 0.880, respectively, which revealed low diagnostic values for AFB microscopy (>0.5 to <0.7) and moderate diagnostic value for both Xpert MTB/RIF and MTBDR*plus* assays (≥0.7 to <0.9). On the other hand, the AUC of culture, AFB microscopy, Xpert MTB/RIF, and MTBDR*plus* assays against CRS were 0.863, 0.626, 0.928, and 0.870, respectively, which demonstrated the moderate diagnostic value of culture (≥0.7 to <0.9), low diagnostic value for AFB microscopy (>0.5 to <0.7), high diagnostic value for Xpert MTB/RIF (≥0.9), and moderate diagnostic value for both Xpert MTB/RIF and MTBDR*plus* assays (≥0.7 to <0.9), respectively, in the detection of EPTB.

The diagnostic performance of these four methods varied across the EPTB specimen when compared with CRS. Both the molecular methods (Xpert MTB/RIF and MTBDR*plus* assay) showed considerable sensitivity for tissue (94.1% and 85.3%, respectively) and pus (95.5% and 93.9%, respectively) specimens compared to the conventional culture (58.8% and 14.7%, respectively) and AFB microscopy (77.3% and 47.0%, respectively) methods. On the contrary, the sensitivities of all the methods were comparatively lower for body fluid specimens like ascitic fluid and pleural fluid. Several previous studies also found poor sensitivity in such fluid specimens for culture methods, while it had higher sensitivity for molecular methods, suggesting tissue biopsies instead of fluids as the effective sample of choice for the detection of EPTB [30,31]. For CSF, culture had a considerably higher sensitivity (84.6%) compared to Xpert MTB/RIF (69.2%) and MTBDR*plus* assay (61.5%) in contrast to CRS. Although it is not attempted in this study, an investigation by Bahr, N.C. et al. demonstrated a considerable increase in the sensitivity of Xpert MTB/RIF assay using CSF pellet after centrifugation [32].

As the paucibacillary nature of EPTB specimens is an established hypothesis, we investigated the grading of AFB microscopy and Xpert MTB/RIF assay to investigate the bacterial load in the specimens. We found that 79.6% (n = 39) of AFB-positive cases were either “scanty” or “1+”, while 78.3% (n = 130) of Xpert MTB/RIF-positive cases were of “very low” and “low” burden. All the scanty/1+ AFB-positive cases were also detected by Xpert MTB/RIF, whereas approximately 85% (n = 110) of low/very low cases were missed in AFB microscopy. However, the newly developed next-generation Xpert MTB/RIF Ultra (Xpert Ultra) assay has an 8 times improved limit of detection (LOD; 16 CFU/mL vs. 114 CFU/mL of Xpert MTB/RIF) and has better sensitivity and specificity than its predecessor [33]. Though this method was not attempted in the current study, this improved version could be more efficient and beneficial for specimens with a low level of bacteria, especially in paucibacillary EPTB specimens [5,34].

Additionally, the two molecular methods (Xpert MTB/RIF and MTBDR*plus* assay) demonstrated 100% sensitivity and specificity for the detection of RIF susceptibility among them. A key reason behind this very high performance was that most of the isolates were RIF-sensitive, and only a few (n = 2) were RIF-resistant. Our previous study with PTB specimens revealed 91.3% agreement in probe hybridization for RIF resistance determination among these two methods [35]. Thus, if there were a greater number of RIF-resistant specimens among these EPTB cases, the scenario might have been different.

Our study has a few limitations. Firstly, we conducted this study in four tertiary care hospitals, which might not reflect the actual scenario of EPTB cases in the community. Secondly, we collected diverse types of specimens which were not equally distributed in numbers. This might hamper the definite interpretation of the test results for each category of the specimens. Lastly, we could not attempt liquid culture and Xpert MTB/RIF Ultra assay in this investigation.

## 5. Conclusions

In this study, when culture and CRS were used as reference standards, Xpert MTB/RIF had better diagnostic performance characteristics than any other methods. To ensure proper and earlier bacteriological diagnosis and drug susceptibility of EPTB cases, molecular methods could be a frontline technique. Additionally, better accuracy and sensitivity of Xpert MTB/RIF, along with superior performance on microscopy negative specimens, warrant the use of Xpert MTB/RIF for early detection, and routine EPTB testing will benefit the TB control programs.

## Figures and Tables

**Figure 1 microorganisms-11-01066-f001:**
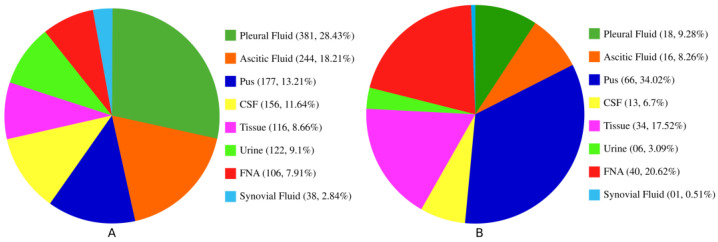
Distribution of specimen types (**A**) and the number of positive cases (**B**) among the enrolled EPTB specimens.

**Figure 2 microorganisms-11-01066-f002:**
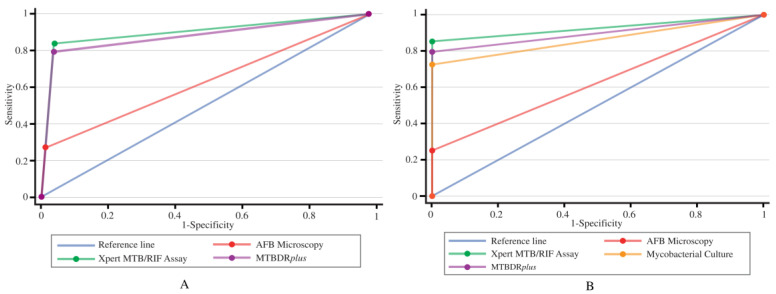
ROC curve analysis for the four TB diagnostic methods. (**A**) The AUC for AFB microscopy, Xpert MTB/RIF, and MTBDRplus when compared with the culture method. (**B**) The AUC for culture, AFB microscopy, Xpert MTB/RIF, and MTBDRplus when compared with CRS.

**Figure 3 microorganisms-11-01066-f003:**
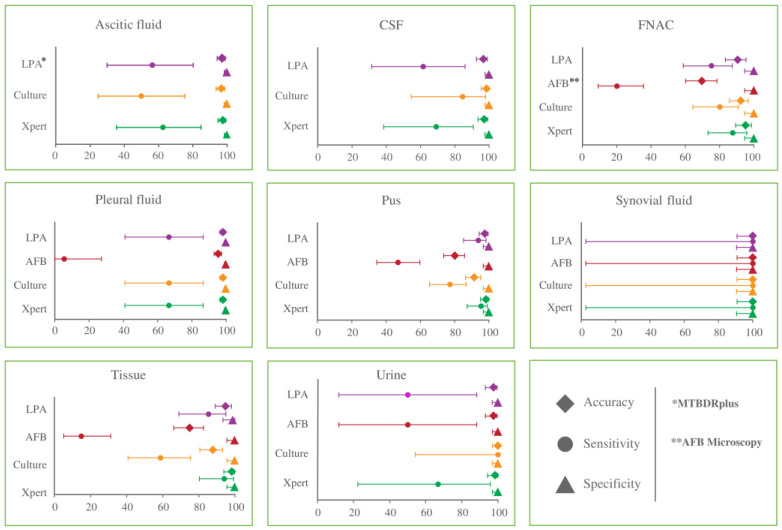
Diagnostic performance of four laboratory methods compared to CRS for EPTB detection in different specimen types. The bar represents a 95% confidence interval of three different diagnostic parameters: sensitivity, specificity, and accuracy.

**Table 1 microorganisms-11-01066-t001:** Diagnostic performance of AFB microscopy, Xpert MTB/RIF t, and MTBDRplus for detection of EPTB cases compared to culture method.

Tests	Results	Culture		Sensitivity (95% CI)	Specificity (95% CI)	PPV (%)	NPV (%)	Accuracy (%)	Kappa Coefficient	*p*-Value	AUC
		Positive n = 141 (10.52)	Negative n = 1199 (89.48%)								
AFB Microscopy	Positive n = 49 (3.66%)	38	11	27.0 (19.8–35.1)	99.1 (98.4–99.5)	77.6	92	91.4	0.3656	<0.001	0.630
	Negative n = 1291 (96.34%)	103	1188								
Xpert MTB/RIF assay	Positive n = 166 (12.39%)	118	48	83.7 (76.5–89.4)	96.0 (94.7–97.0)	71.1	98	94.7	0.739	<0.001	0.898
	Negative n = 1174 (87.61%)	23	1151								
MTBDR*plus* assay	Positive n = 154 (11.49%)	112	42	79.4 (71.8–85.8)	96.5 (95.3–97.5)	72.7	97.6	94.7	0.7296	<0.001	0.880
	Negative n = 1186 (88.51%)	29	1157								

**Table 2 microorganisms-11-01066-t002:** Diagnostic performances of culture, AFB microscopy, Xpert MTB/RIF, and MTBDRplus for detection of EPTB cases compared to the composite reference standard.

Tests	Results	CRS		Sensitivity (95% CI)	Specificity (95% CI)	PPV (%)	NPV (%)	Accuracy (%)	Kappa Coefficient	*p*-Value	AUC
		Positive n = 194	Negative n = 1146								
Culture	Positive n = 141 (10.52%)	141	0	72.7 (65.8–78.8)	100 (99.7–100)	100	95.6	96	0.8198	<0.001	0.863
	Negative n = 1199 (89.48%)	53	1146								
AFB Microscopy	Positive n = 49 (3.66%)	49	0	25.3 (19.3–32.0)	100 (99.7–100)	100	88.8	89.2	0.3663	<0.001	0.626
	Negative n = 1291 (96.34%)	145	1146								
Xpert MTB/RIF assay	Positive n = 166 (12.39%)	166	0	85.6 (79.8–90.2)	100 (99.7–100)	100	97.6	97.9	0.9102	<0.001	0.928
	Negative n = 1174 (87.61%)	28	1146								
MTBDR*plus* assay	Positive n = 154 (11.49%)	154	0	79.4 (73.0–84.8)	100 (99.7–100)	100	96.6	97	0.8682	<0.001	0.870
	Negative n = 1186 (88.51%)	40	1146								

**Table 3 microorganisms-11-01066-t003:** Burden of MTBC isolates among different EPTB specimens determined by AFB microscopy and Xpert MTB/RIF assay.

Tests	Gradings	Positive Cases n (%)	Specimen Types							
			Pleural Fluid	Ascitic Fluid	Pus	CSF	Tissue	Urine	FNAC	Synovial Fluid
AFB Microscopy Positive (n = 49)	Scanty	21 (42.9)	1	0	10	0	4	1	4	1
	1+	18 (36.7)	0	0	14	0	0	2	2	0
	2+	5 (10.2)	0	0	3	0	0	0	2	0
	3+	5 (10.2)	0	0	4	0	1	0	0	0
Xpert MTB/RIF Positive (n = 166)	Very low	54 (32.5)	4	10	8	5	15	1	11	0
	Low	76 (45.8)	7	0	31	4	15	1	18	0
	Medium	28 (16.9)	1	0	19	0	1	2	4	1
	High	8 (4.8)	0	0	5	0	1	0	2	0

**Table 4 microorganisms-11-01066-t004:** Comparison of Xpert MTB/RIF and GenoType MTBDRplus for the detection of Rifampicin susceptibility.

Tests	Results	Xpert MTB/RIF Assay		Sensitivity (95% CI)	Specificity (95% CI)	PPV (%)	NPV (%)	Accuracy (%)	*p*-Value
		Positive (n = 166)	Negative (n = 1174)						
MTBDRplus assay	Positive (n = 154)	145	9	87.3 (81.3–92.0)	99.2 (98.5–99.6)	94.2	98.2	97.8	<0.001
	Negative (n = 1199)	21	1165						
		Sensitive (n = 143)	Resistant (n = 2)						
	RIF-Sensitive (n = 143)	143	0	100 (97.5–100)	100 (15.8–100)	100	100	100	-
	RIF-Resistant (n = 2)	0	2						

## Data Availability

The data presented in this study are available on request from the corresponding author. The data are not publicly available due to ethical restrictions.

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
