# Peer review of "Diagnostic Performance of Different Laboratory Methods for the Detection of Extrapulmonary Tuberculosis"

_microorganisms, 2023, doi:10.3390/microorganisms11041066_

Round 1

Reviewer 1 Report

The manuscript titled "Diagnostic Performance of Different Laboratory Methods for the Detection of Extrapulmonary Tuberculosis" is an interesting and significant study wherein the authors collected 1340 patient specimen and investigated the sensitivity and specificity of AFB, culture, Xpert and MTBDRplus in detecting M. tuberculosis and RIF resistance. The findings of this study are very important for the clinical community while diagnosing tuberculosis. Overall this manuscript is well drafted and conclusions are clearly written

Minor comments

The language is confusing in some places, I suggest a complete proof reading by a language expert

In some places, the authors use non technical vocabularies such as Indoor and outdoor in following sentence. Page 2,Line 96-97L: either visited in 96 outdoor or were admitted in indoor departments. This may be rewritten as In-patients and Out-patients. 

In methods, the authors say that they collected 1300 specimen, but in results and abstract it is mentioned as 1340 specimen

Reviewer 2 Report

 The paper by Uddin et al is well written and interesting to read. The overall manuscript presentation was impressive and interesting. Title is clear and informative; it displays the main objective of the study. The literature review follow the specific aim of the study. The figures/tables have almost sufficient, good quality and appropriately illustrative of the paper contents. The study from a scientific point of view seems to be well done and presents good results, from where to derive valid conclusions. Introduction summarizes relevant research to provide context and clearly state the problem. The research methods used ensure the reliability of the obtained results. However I have some minor comments:

Line 36 : change “Mycobacterium tuberculosis (MTB) ” to “Mycobacterium tuberculosis complex (MTBC)” please use MTBC throughout the manuscript.

Lines 49-51: “One of  the major barriers is the difficulty of obtaining specimens from infection sites involves  invasive procedure including surgery.” Is not clear and need rephrasing

Lines 61-62: “The presence of PCR inhibitors in the clinical specimens can compromise the diagnostic performance of this technique.” Please add a reference.

Line 64: “Xpert” is a general name for the automated molecular tests (i.e. Xpert®  MTB/RIF , Xpert® Xpress Flu/RSV & …) developed by Cepheid. Please mention the complete name of the test. please use Xpert®  MTB/RIF throughout the manuscript.

Figure 2 A & B : Please use the same colour for each method in Figure 2 A & B. The current version is confusing. Why culture is missing in Figure 2 A?

Reviewer 3 Report

Comments:

I appreciate the work done by the authors in their manuscript. They have addressed some important questions and generated a preliminary report on the diagnostic performance of various laboratory methods for the detection of extrapulmonary tuberculosis. The results have been presented in an elegant and succinct manner. The inferences made by the authors are justified and align with their observations and conclusions.

However, I have a few comments and one edit to mention:

1.       In the abstract and line 107, please correct the number of samples to 1340.

2.       In line 60: There are also several non-invasive methods reported for the diagnosis of EPTB, including stool PCR, as mentioned in the study with PMID 32345991. Please consider including this information.

3.       In line 150: Please clarify why the numbers do not match with those in section 3.2.

4.       In Figure 3: The variability in the data is challenging to comprehend using the dot plot. I recommend using a box plot instead, as it provides a clearer representation of the first quartile (Q1), second quartile (Q2), and third quartile (Q3), as well as any outliers contributing to the high variability in some of your datasets.
